# Differentiation of Cytopathic Effects (CPE) induced by influenza virus infection using deep Convolutional Neural Networks (CNN)

**Ting-En Wang**[1], **Tai-Ling Chao**[2], **Hsin-Tsuen Tsai**[2], **Pi-Han Lin**[2], **Yen-Lung Tsai**[1]*, **Sui-Yuan Chang**[2,3]*

**1** Department of Mathematical Sciences, National Chengchi University, **2** Department of Clinical Laboratory Sciences and Medical Biotechnology, National Taiwan University College of Medicine, Taipei, Taiwan, **3** Department of Laboratory Medicine, National Taiwan University Hospital, Taipei, Taiwan

* yenlung@nccu.edu.tw (YLT); sychang@ntu.edu.tw (SYC)

## Abstract

Cell culture remains as the golden standard for primary isolation of viruses in clinical specimens. In the current practice, researchers have to recognize the cytopathic effects (CPE) induced by virus infection and subsequently use virus-specific monoclonal antibody to confirm the presence of virus. Considering the broad applications of neural network in various fields, we aimed to utilize convolutional neural networks (CNN) to shorten the timing required for CPE identification and to improve the assay sensitivity. Based on the characteristics of influenza-induced CPE, a CNN model with larger sizes of filters and max-pooling kernels was constructed in the absence of transfer learning. A total of 601 images from mock-infected and influenza-infected MDCK cells were used to train the model. The performance of the model was tested by using extra 400 images and the percentage of correct recognition was 99.75%. To further examine the limit of our model in evaluating the changes of CPE overtime, additional 1190 images from a new experiment were used and the recognition rates at 16 hour (hr), 28 hr, and 40 hr post virus infection were 71.80%, 98.25%, and 87.46%, respectively. The specificity of our model, examined by images of MDCK cells infected by six other non-influenza viruses, was 100%. Hence, a simple CNN model was established to enhance the identification of influenza virus in clinical practice.

## Author summary

Observation of cytopathic effects (CPE) induced by virus infection is a practical method to determine the prsence of viruses in the clinical specimens. However, CPE observation is labor-intensive and time-consuming because it requires medical examiner to inspect cell morphology changes for a period of time. Here, Convolutional Neural Networks (CNN) was applied to improve the disadvantage of CPE observation by using influenza virus as an example. To reduce the requirement for large image input of every clinical test, small amount of data was used to train our CNNs model without transfer learning and the trained model was examined with testing image data taken at 25hr post virus infection. The recognition of testing data shows that the model can identify CPE at 25hr and the

**Data Availability Statement:** All relevant data are within the manuscript and its Supporting Information files.

**Funding:** The funders had no role in study design, data collection and analysis, decision to publish, or preparation of the manuscript.

**Competing interests:** The authors have declared that no competing interests exist.

high specificity of the model can differentiate the CPE induced by influenza viruses from those by other non-influenza viruses. The limit of our model was further examined by more experimental data of influenza-induced and mock-infected images, and the result shows our model can detect the slight changes at the initial stage of CPE development. Hence, our deep CNN model can significantly shorten the timing required to identify virus-induced cytopathic effects.

## Introduction

Despite the availability of rapid tests and nucleic acid amplification assays for quick identification of virus infection, isolation of viruses by the cell culture system remains as one of the golden standards for identifying virus pathogens, especially for emerging virus species. Nevertheless, observation of cytopathic effects (CPE) induced by virus infection is relatively subjective, and requires subsequent reagents such as virus-specific monoclonal antibody to confirm the presence of virus [1,2]. Moreover, it will take longer for cytopathic effects to develop if the amounts of viruses in the inoculated specimens are insufficient or due to some virus strain-specific effects [2,3], which makes the observation of cytopathic effects quite labor-intensive. A better and more objective way to identify cytopathic effects is required.

Nowadays, numerous medical tasks have utilized neural networks to solve the problems or to get better solutions because many tasks we desire to solve were hardly solved by traditional stochastic methods [4–7]. In particular, convolutional neural network (CNN) is a remarkably appropriate model for image recognition and it can differentiate the differences of many classifications at the expert level [5,6,8]. The two special layers, convolutional layer and pooling layer are inspired from the visual system. The convolutional layer of CNN model extracts features from the previous layer and the extracted features are more complex in the later layer as visual mechanism [9,10]. Another important property of the CNN model is the independence of object location for image-recognition [11]. This characteristic is essential for recognition of cytopathic effects because it seems impossible to control the location of cytopathic effects upon virus infection. Therefore, we plan to utilize the properties of the CNN model to solve the challenges in specific identification of cytopathic effects.

In this study, a deep CNN model was developed to recognize the cytopathic effects induced by influenza virus infection of a routinely used cell line, Madin-Darby canine kidney (MDCK) cells. Influenza virus is known to cause annual outbreak of respiratory illness worldwide in people of all ages, and it is one of the most important respiratory viruses causing medically attended acute respiratory illness [12]. The MDCK cell line, first reported in 1966 [13], is the most routinely used cell line to inspect cytopathic effects induced by influenza viruses [1,2]. In this study, we used different input of influenza virus to infect MDCK cells and took photos of cell morphology changes at different time points in the hope to train the CNN model to recognize cytopathic effects induced by influenza virus infection. A total of 601 photos were collected with various time points and viral titers to train the CNN model, and another 400 photos were used to test whether this model could significantly differentiate the cytopathic effects from the normal cell morphology. Additionally, 1190 photos were obtained from a new experiment to test the limit of the model.

## Results

### Model training

Influenza-induced cytopathic effect is a continuous process, beginning with mild morphological changes within the MDCK cells, and then accumulating over time. Usually, the infection

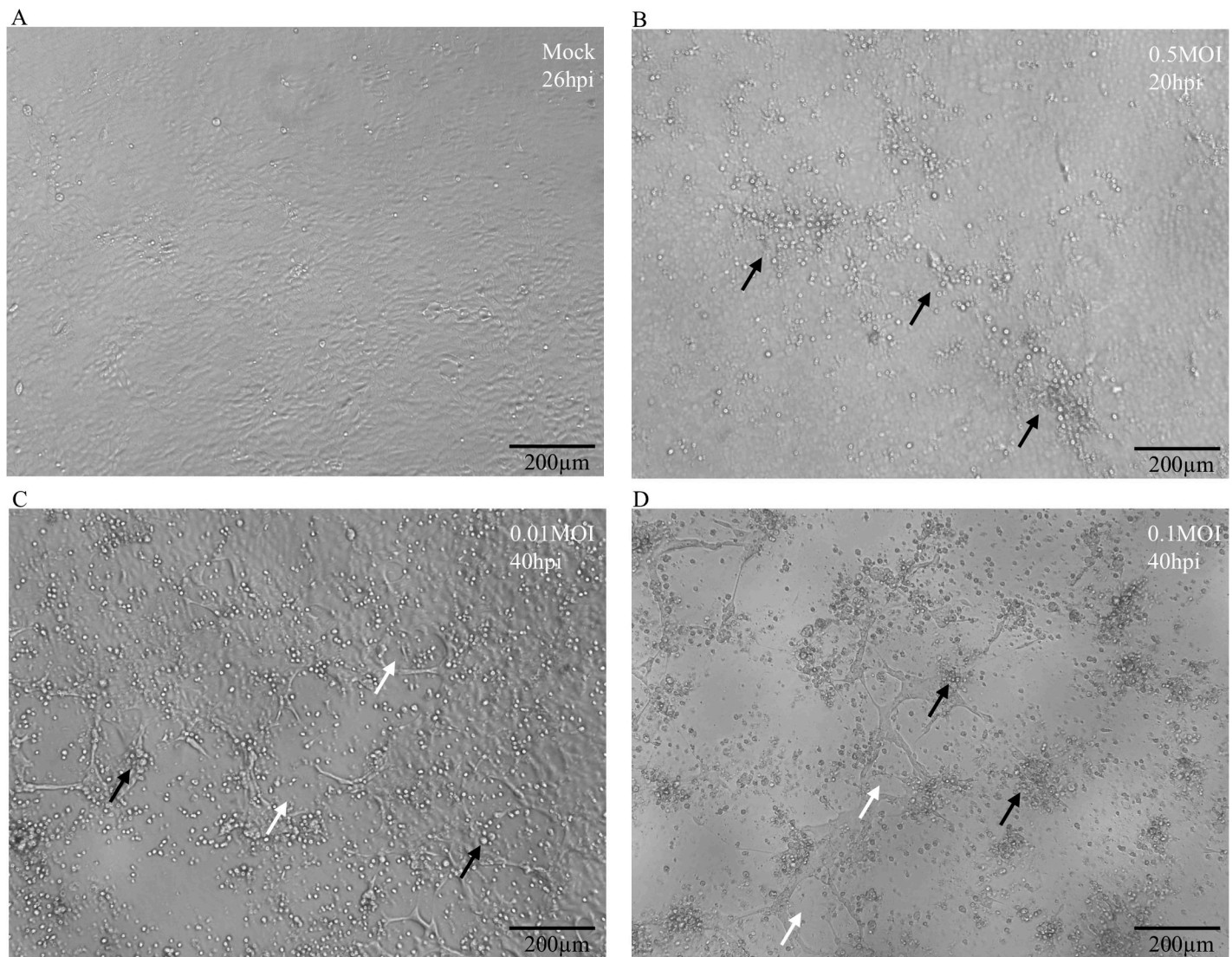

**Fig 1. Influenza-induced cytopathic effects (CPEs) in MDCK cells.** MDCK cells were mock-infected (a), or infected with influenza viruses, which resulting in 20% (b), 50% (c) and 80% (d) of cytopathic effects. Scale bars: 200 μm. Black arrows indicate influenza-induced cytopathic effects. White arrows indicate the area where all infected cells were completely detached and floating in the culture medium.

process is described by using the percentage of cytopathic effects observed overtimes, and different inputs of influenza viruses could induce different levels of cytopathic effects. The mock-infected MDCK cells are tightly adjacent to each other as shown in Fig 1A. In the initial development of cytopathic effects, the MDCK cells become smaller and rounded as shown in Fig 1B. At this stage, only 20% of the cells developed cytopathic effects. In the Fig 1C, 50% of the cells became rounded and some of them began to be detached from the culture flask due to cell death induced by severe cytopathic effects. At this stage, the influenza-induced cytopathic effects are obvious for a medical technologist to recognize under the microscope. When more cells were infected by influenza viruses as shown in Fig 1D, 80% of the cells had cytopathic effects or had been detached from the culture flask. Photographs were taken at different time points to collect various phases of cytopathic effects. The numbers of photographs collected consist of virus-infected cells with different virus input and were summarized in Table 1. The

**Table 1. Information table for the Training 1 training data.**

| | Condition | | Sample numbers |
|---|---|---|---|
| | Dose (M.O.I.) | Time (hpi) | |
| Negative samples | - | 20 | 36 |
| | - | 24 | 48 |
| | - | 26 | 50 |
| | - | 40 | 20 |
| Positive samples | 0.5 | 20 | 46 |
| | 0.1 | 24 | 15 |
| | 0.5 | 24 | 34 |
| | 0.5 | 26 | 55 |
| | 1 | 26 | 55 |
| | 2 | 26 | 54 |
| | 0.1 | 27 | 7 |
| | 0.01 | 40 | 145 |
| | 0.1 | 40 | 36 |

Abbreviation: M.O.I. (multiplicity of infection); hpi (hours post infection)

intervals between the groups of photographs were not regular because we assumed the model would be more accurate with a more diverse training data. Examples of different levels of influenza-induced cytopathic effects and mock-infected MDCK cells used in our dataset were shown in Fig 1. A total of 601 photos were utilized to train the Training 1 and amongst them, 154 were mock-infected control images.

The original images were colored, but were converted into grayscale pictures to be inputted into our model. Reason being, if a colored photo with a size 1024x1360 pixels were used, the dimension for input would require three channels, red, blue and green color which is three times as much as the dimensions needed with a grayscale image. Therefore, we converted all colored photos into grayscale photos to reduce the input dimensions. Furthermore, we considered the property of recognition where identification of cytopathic effects, theoretically, relied on the patterns of changes instead of color variance. Third, Bui *et al.* demonstrated that the use of grayscale pictures can generate better accuracy in differentiating figures in some neural network models than color ones. [14]. Therefore, grayscale photos were chosen as the input of the model, considering it may reduce the potential interference caused by colors and enhance the accuracy.

## Model validation

After training the Training 1 with 1200 epochs, 99% of the training data (including the training set and validation set) can be recognized accurately as tabulated in Table 2. For examination of overfitting, we further took 400 images from an independent experiment, including 300 images of infected cells and 100 images of uninfected cells as shown in Fig 2A–2C and S1 Table. Our trained model identified 99.75% of the testing data. Moreover, all of the 300 positive pictures were identified correctly, with no error, resulting in a 100% negative predictive value. The accuracy for negative samples was also high with a percentage of 99 and the positive predictive value was 99.66% (Table 2).

## Specificity of the model

In the clinical practice, MDCK cells are regularly used to isolate influenza viruses. However, influenza viruses are not the only species that can infect MDCK cells. Other viruses, such as

**Table 2. Comparison of Training 1 and Training 2 with 1200 epochs weights and the saved weights.**

| | Training 1 | | Training 2 | |
|---|---|---|---|---|
| Weight chosen | 1200 epochs | saved weights | 1200 epochs | saved weights |
| Training numbers | 601 | 601 | 503 | 503 |
| Training accuracy | 0.9900 | 0.9600 | 0.9860 | 0.9662 |
| Testing numbers | 400 | 400 | 498 | 498 |
| Testing accuracy | 0.9975* | 0.9925 | 0.9457 | 0.9216 |
| PPV [a] | 0.9966* | 0.9900 | 0.9673 | 0.9278 |
| NPV [b] | 1** | 1 | 0.8846 | 0.9 |

Abbreviation: Pos, positive samples; Neg, negative samples; PPV, positive predictive value; NPV, negative predictive value.

[a] PPV = numbers of true positives/(numbers of true positives+ numbers of false positives). The ideal value of the PPV for a perfect test will be 1, while the worst value will be zero.

[b] NPV = numbers of true negatives/(numbers of true negatives+ numbers of false negatives). The ideal value of the NPV for a perfect test will be 1, while the worst value will be zero

**: p-value <0.01

*: p-value <0.05

RSV and HSV, have been described to infect MDCK cells and induce distinct patterns of cytopathic effects [15]. Therefore, we would like to verify whether our model would miss-identify other virus-induced cytopathic effects as influenza infected samples. Additional experiment images of MDCK cells infected with adenovirus, coxsackievirus B3, HSV-1, HSV-2, parainfluenza virus, and RSV were included for analysis. Twenty images were taken for each virus infected cells at 40 hours post infection, considering 40 hours being the last time point of observation for influenza-infection in this study. HSV-1 formed cytopathic effects as reported by Meguro *et al*. [15] (Fig 3A). On the contrary, HSV-2 did not induce any cytopathic effects in MDCK cells as shown in Fig 3B. Adenovirus induced a relatively minor changes in cell morphology compared to mock-infected MDCK cells in Fig 3C. No cytopathic effect was observed in MDCK cells infected by other viruses (Fig 3D–3F). Next, the images collected from these non-influenza viruses infected MDCK cells were included for analysis to examine the specificity of our model. All of the images mentioned above were not recognized as influenza-infected MDCK cells. In addition, twenty images were taken for each virus infected cells at 16, 25, and 28 hpi and validated by the model. The results were consistent with those using the 40 hpi images (S3 Table). These tests further validate the specificity of our model towards influenza infection.

## Test for earlier and later images

According to our preliminary analysis on the testing set, this model had a considerably high accuracy when differentiating different virus input, 0.05 multiplicity of infection (M.O.I.) and 0.5 M.O.I., at 25 hours post infection (hpi) (Table 2). Next, we aimed to examine if our model could differentiate influenza-induced cytopathic effects at earlier time points after infection. We fed more experiment data taken at earlier time points, 16 hpi, into our trained model (Fig 2D–2F). We also collected images at 28 and 40 hpi, as shown in Fig 2G and 2H, to examine the recognition rate for late phase of cytopathic effects by our model. The numbers of the influenza experiment data collected at different time points were tabulated in Table 3. Our model had great accuracy identifying all negative samples at 16, 28, and 40 hpi with accuracy of 97.14%, 99.18%, and 99.02%, respectively (Table 4). The positive samples taken at 28 hpi were also recognized specifically with accuracy of 97.83%. However, the recognition rate of images taken at

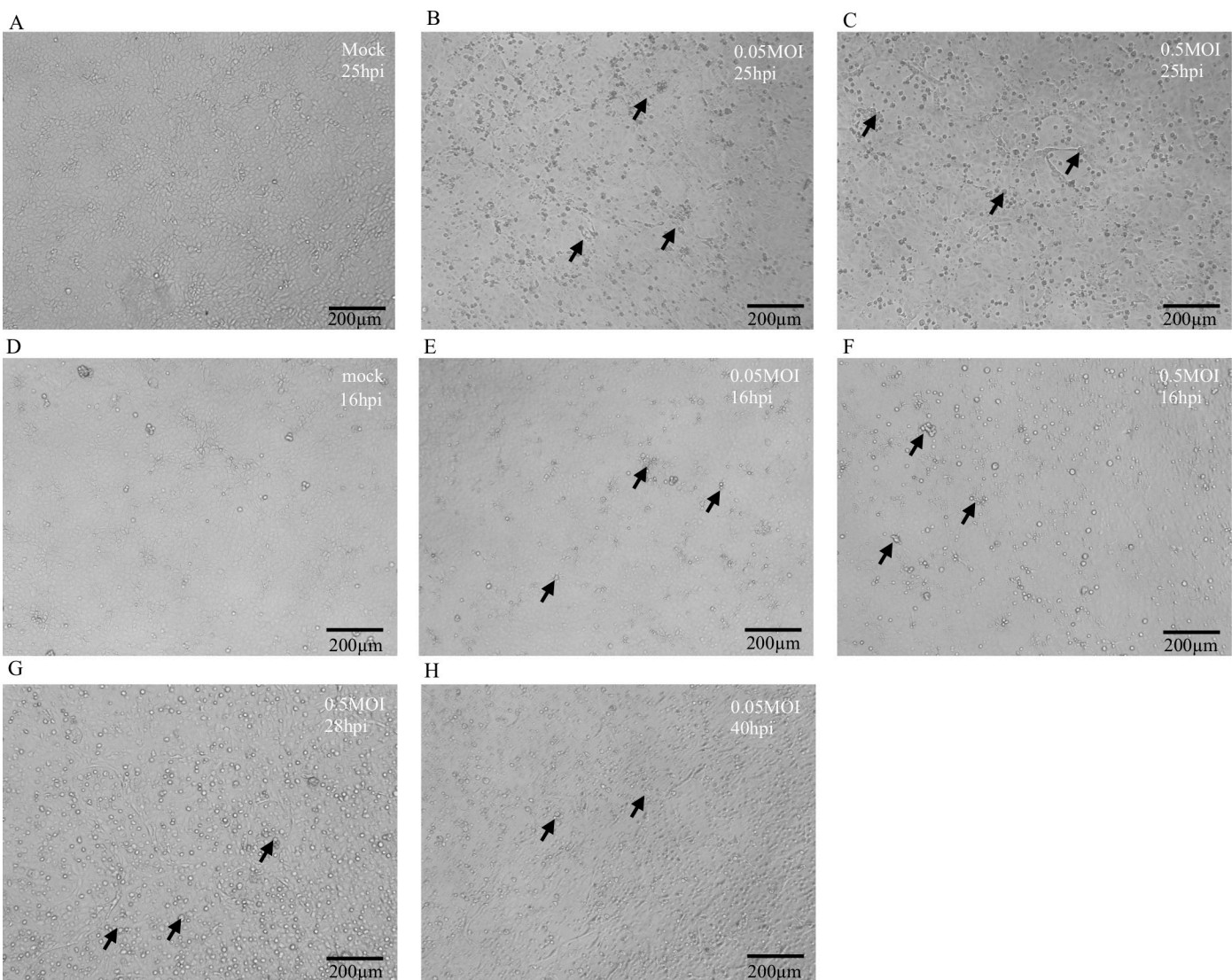

**Fig 2. Influenza-induced cytopathic effects (CPEs) at different time points after infection.** Images of MDCK cells were taken at 25hpi (hours post infection) (a-c), 16hpi (d-f), 28hpi (g), and 40hpi (h). (a) and (d) represent MDCK cells which were mock-infected. (b)(e) and (c)(f) represent MDCK cells which were infected with influenza viruses at 0.05 M.O.I. and 0.5 M.O.I., respectively. (g) represents image of MDCK cells infected with 0.5 M.O.I. of influenza virus at 28hpi. (h) represents image of MDCK cells infected with 0.05 M.O.I. of influenza virus at 40hpi. Scale bars: 200 μm. Black arrows indicate influenza-induced cytopathic effects.

40 hpi was slightly reduced to 83.08%, probably due to the detachment of virus-infected cells at later time points. The recognition rate of the earlier experiment data (16 hpi) was not as expected, with the accuracy of 88.88% and 15.92% for positive samples infected at 0.5 M.O.I. and 0.05 M.O.I., respectively. The reason for deficiency in the recognition of those pictures at 0.05 M.O.I., could be attributed to the fact that the morphology of MDCK cells infected with influenza in a shorter period was too similar to that of the mock-infected cells (Fig 2D–2F). However, no significant difference could be observed even by manual identification at this stage, either.

The positive and negative predictive values of these experiment data were also determined. The positive predictive values for the positive samples in the influenza experiment set were 97.59%, 99.63%, and 99.55% at 16, 28, and 40 hpi, respectively. The positive experiment images taken at 28 hpi had a better recognition rate and the negative predictive value was 95.31%. On

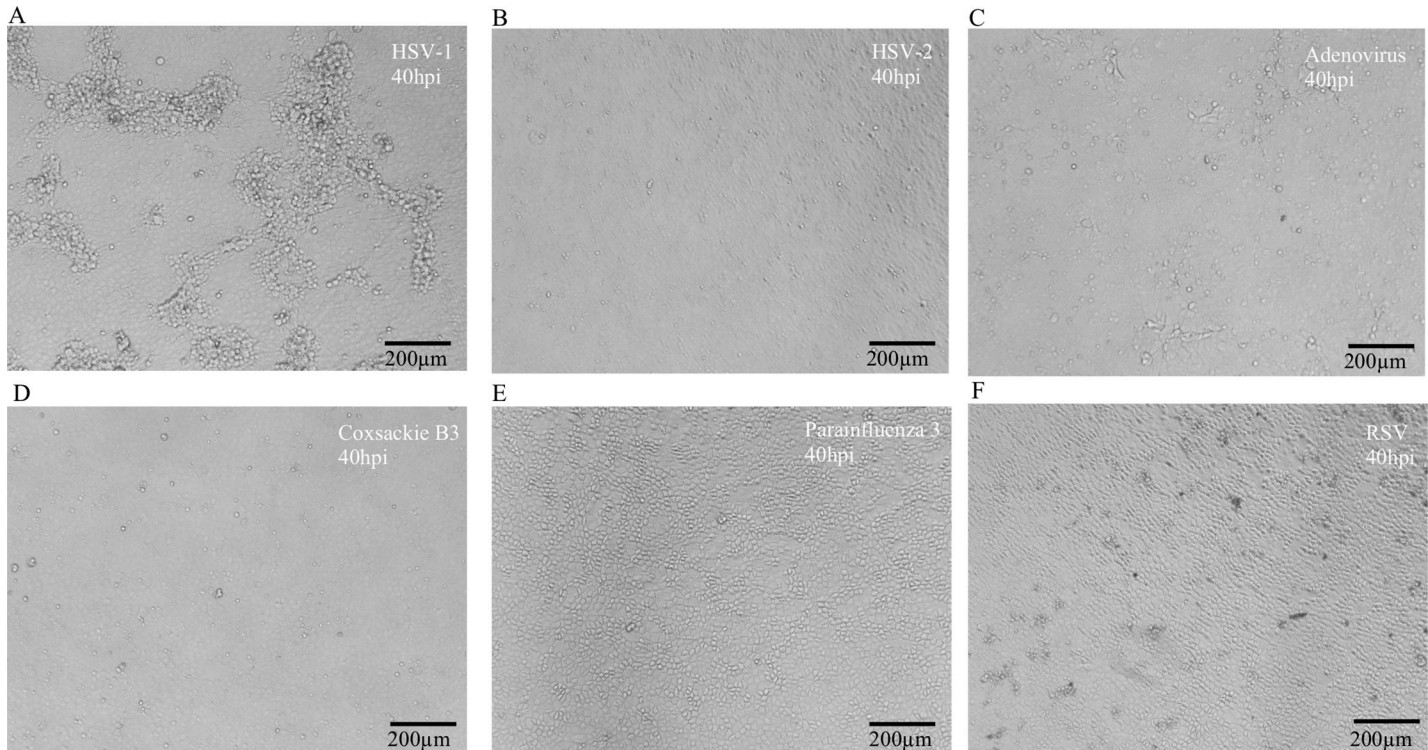

**Fig 3. Infection of MDCK cells by non-influenza viruses.** MDCK cells were infected with herpes simplex virus type 1 (HSV-1) (a), herpes simplex virus type 2 (HSV-2) (b), adenovirus (c), coxsackie B3 virus (d), parainfluenza 3 virus (e), and respiratory syncytium virus (RSV)(f). The images were taken at 40 hpi (hours post infection). Scale bars: 200 μm.

the other hand, due to the lower sensitivity of initial and late stages of influenza-induced cytopathic effects, the negative predictive values were not satisfactory, where merely 54.61% for 16 hpi experiment data and 68.91% for 40 hpi experiment data. When the machine recognized the influenza experiment data, the positive and negative predictive values were 99.09% and 68.57%, respectively. Finally, the positive and negative predictive values of the machine for all of the influenza experiment data and testing data were 99.27% and 73.55%, individually as tabulated in Table 4.

**Table 3. Information table for the influenza experiment data set.**

|  | Infection condition | | Sample numbers |
| --- | --- | --- | --- |
|  | Dose (M.O.I.) | Time (hpi) |  |
| Negative samples | - | 16 | 140 |
|  | - | 28 | 123 |
|  | - | 40 | 103 |
| Positive samples | 0.05 | 16 | 113 |
|  | 0.5 | 16 | 162 |
|  | 0.05 | 28 | 154 |
|  | 0.5 | 28 | 123 |
|  | 0.05 | 40 | 150 |
|  | 0.5 | 40 | 122 |

Abbreviation: M.O.I. (multiplicity of infection); hpi (hours post infection)

**Table 4. Comparison of Training 1 and Training 2 with 1200 epochs weights and the saved weights on Experiment Data.**

| Weight chosen | | Training 1 | | Training 2 | |
|---|---|---|---|---|---|
| | | 1200 epochs | saved weights | 1200 epochs | saved weights |
| **Accuracy of additional testing data** | 16 hpi Pos+Neg[c] (415)[d] | 0.718** | 0.8024 | 0.6168 | 0.7638 |
| | 16 hpi Neg (140) | 0.9714 | 0.9214 | 1 | 0.9285 |
| | 16 hpi Pos (275) | 0.5890*** | 0.7418 | 0.4218 | 0.6800 |
| | 16 hpi 0.5M.O.I. (162) | 0.8888*** | 0.9815 | 0.6790 | 0.9320 |
| | 16 hpi 0.05M.O.I. (113) | 0.1592* | 0.3982 | 0.0530 | 0.3185 |
| | 28 hpi Pos+Neg (400) | 0.9825 | 0.9625 | 0.9625 | 0.98 |
| | 28 hpi Neg (123) | 0.9918 | 0.9186 | 1 | 0.9349 |
| | 28 hpi Pos (277) | 0.9783 | 0.9819 | 0.9458 | 1 |
| | 40 hpi Pos+Neg (375) | 0.8746*** | 0.9120 | 0.7733 | 0.9733 |
| | 40 hpi Neg (103) | 0.9902 | 0.9417 | 1 | 0.9708 |
| | 40 hpi Pos (272) | 0.8308*** | 0.9007 | 0.6875 | 0.9742 |
| **Accuracy of other viruses data** | HSV-1 | 1 | 0.95 | 1 | 0.95 |
| | HSV-2 | 1 | 1 | 1 | 1 |
| | RSV | 1 | 1 | 1 | 0.35 |
| | Parainfluenza virus | 1 | 0.95 | 1 | 0.7 |
| | Coxsackievirus B3 | 1 | 1 | 1 | 1 |
| | Adenovirus | 1 | 0.9 | 0.95 | 0.8 |
| **PPV[a]** | 16 hpi | 0.9759 | 0.9488 | 1 | 0.9492 |
| | 28 hpi | 0.9963 | 0.9645 | 1 | 0.9719 |
| | 40 hpi | 0.9955 | 0.9760 | 1 | 0.9888 |
| | 16 hpi+28 hpi+40 hpi | 0.9909 | 0.9639 | 1 | 0.972 |
| | All testing data | 0.9927 | 0.9714 | 0.9871 | 0.9569 |
| **NPV[b]** | 16 hpi | 0.5461 | 0.645 | 0.4682 | 0.5963 |
| | 28 hpi | 0.9531 | 0.9576 | 0.8913 | 1 |
| | 40 hpi | 0.6891* | 0.7822 | 0.5478 | 0.9345 |
| | 16 hpi+28 hpi+40 hpi | 0.6857*** | 0.7669 | 0.5856 | 0.7840 |
| | All testing data | 0.7355*** | 0.8089 | 0.6370 | 0.8072 |

Abbreviation: Pos, positive samples; Neg, negative samples; PPV, positive predictive value; NPV, negative predictive value; HSV-1, herpes simplex virus type 1; HSV-2, herpes simplex virus type 2; RSV, respiratory syncytium virus

[a] PPV = numbers of true positives/(numbers of true positives+ numbers of false positives). The ideal value of the PPV for a perfect test will be 1, while the worst value will be zero.

[b] NPV = numbers of true negatives/(numbers of true negatives+ numbers of false negatives). The ideal value of the NPV for a perfect test will be 1, while the worst value will be zero

[c] Pos+Neg: positive samples and negative samples

[d] Numbers in the brackets represent the amount of photos

***: p-value <0.001

**: p-value <0.01

*: p-value <0.05

## Training the model by reassigned training data

Since the virus titers might vary in the clinical specimens, we reassigned the original training data and the testing data. All of 1001 pictures were randomly divided into two groups, the training and testing materials. To avoid uneven distribution of the images in the training data and testing data, we split each image set equally, and the amount of images in the reassigned training data and testing data were 503 and 498, respectively. The structure of the model and

the training epochs were not changed. We named the previous training as "Training 1" and the retrained training as "Training 2". The differences of training and testing data of these two trainings and the analysis results were summarized in Fig 4.

### Examination of Training 2

The accuracy of the resigned training data of Training 2 was 98.6%. Next, the resigned testing data was used to examine whether Training 2 was overfitting the training data. The accuracy of the testing data was 94.57%, slightly lower than that of the training data. The influenza experiment data were also tested and the results were compared with Training 1 as tabulated in Table 4. The accuracy of the influenza experiment data for 16 hpi (61.68% v.s. 71.8%), 28 hpi (96.25% v.s. 98.25%), and 40 hpi (77.33% v.s. 87.46%) was all slightly lower than those of Training 1, as shown in Table 4. However, the specificity of the influenza experiment data by Training 2 was slightly better than those of Training 1 for 16 hpi (100% v.s. 97.14%), 28 hpi (100% v.s. 99.18%), and 40 hpi (100% v.s. 99.02%), respectively. Finally, we used the same images of MDCK cells infected with adenovirus, coxsackievirus B3, HSV-1, HSV-2, parainfluenza virus, and RSV to examine whether Training 2 can differentiate these non-influenza-induced CPEs from those by influenza viruses. Most of the classification results of Training 2 were perfect without any image being misjudged as influenza-infected cells, except one image from adenovirus-infected cells (5%). (Table 4).

### Comparison of Training 1, Training 2, and visual examination

There are minor differences in the performance between Training 1 and Training 2, regarding the accuracy, specificity, positive predict values, and negative predict values. The chi-square analysis was performed to determine whether the difference is statistically significant. The testing accuracy and negative predict value of the testing data of Training 1 are significantly higher than those of Training 2 (Table 2).

Moreover, Training 1 showed a significant higher recognition rate on earlier recognition than those of Training 2, except the accuracy of 16 hpi negative images of influenza experiment set, which did not achieve statistically significance. Nevertheless, the recognition of the late stage by Training 1 was significantly higher than those of Training 2.

To examine our model can reduce the time of identifying influenza-infected MDCK cells, we compared our model with visual examination. As shown in Table 5, both Training 1 and Training 2 performed significantly higher accuracy than visual examination. The significantly higher value of negative predictive value of Training 1 and Training 2 implicated that our model can recognize influenza-induced CPE images earlier than human.

### Result of other weights

In the beginning stage of model training, we split 20% of training data as validation dataset. We saved the weights to examine if the accuracy of the validation data would be better than before. We also examined all of the testing data accuracy with those saved weights, but the amount of saved weights was numerous. Thus, we only tried the weights with validation set accuracies above 90%. The results of those weights from two Trainings were almost as great as the weights after 1200 epochs training, as shown in Tables 2 and 4. However, these results were slightly worse than the final weights after 1200 epoch training. Those saved weights were saved within 100 epoch. It probably implied that the model was convergent quickly in the beginning, but 1200 epochs were necessary.

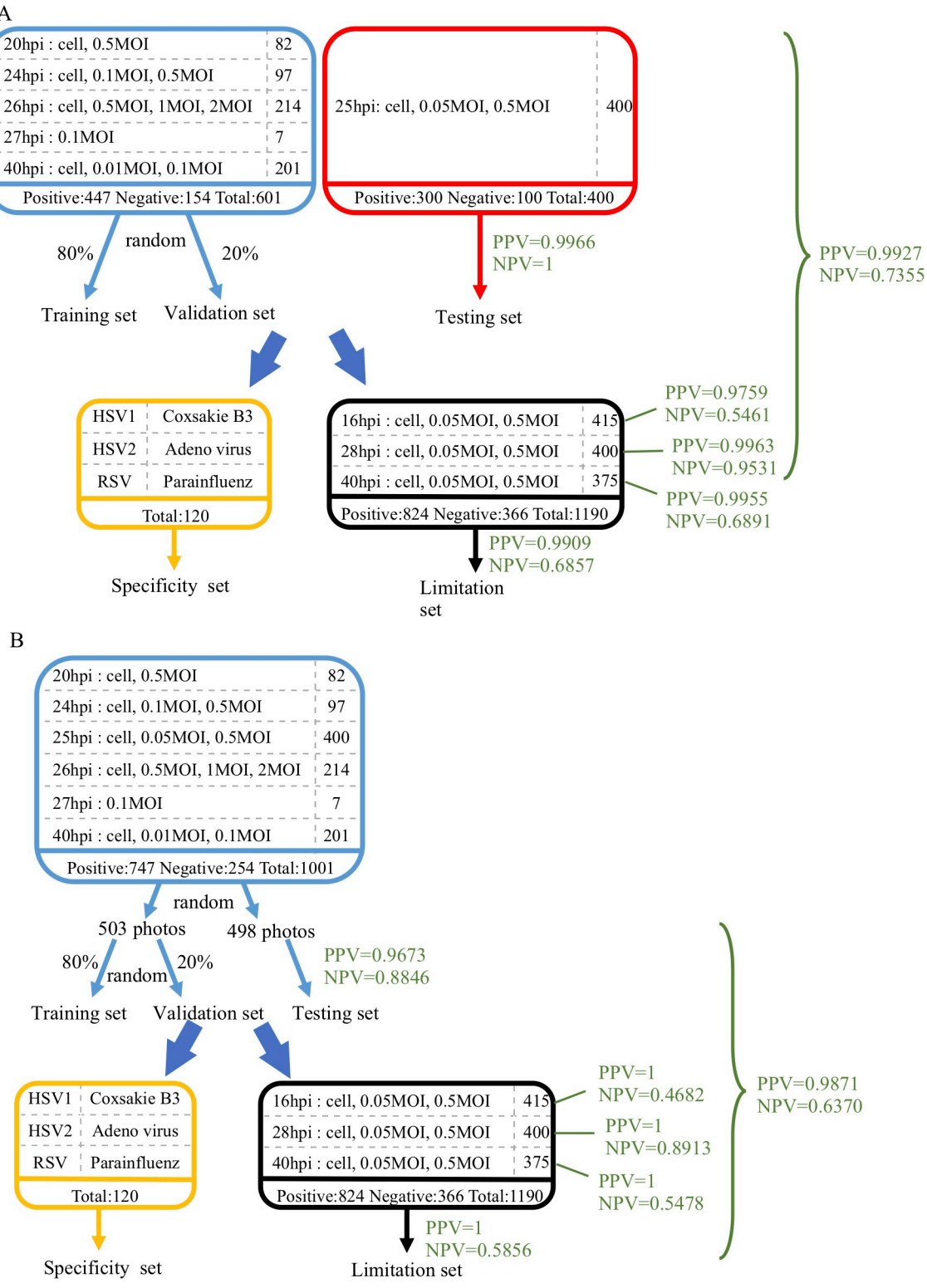

**Fig 4. Flow charts for Training 1 A and Training 2 B. (A) The flow chart of Training 1.** The blue, red, and black frames indicate the data condition of influenza infection in the training process, testing process, and limitation set, respectively. The viruses in the yellow frame were used for specificity test. **(B) The flow chart of Training 2.** The blue frame indicates the data condition of influenza infection in the training and testing process. The viruses in the yellow frame were used for specificity test. The black frame indicates the data condition of influenza infection in the limitation set.

**Table 5. Comparison of Training 1, Training 2 and visual examination.**

| | Accuracy | PPV [a] | NPV [b] | confusion matrix | | |
|---|---|---|---|---|---|---|
| | | | | Actual | Predicted | |
| | | | | | Pos | Neg |
| **Training 1** | 0.9975*** | 0.9966* | 1*** | **Pos** | 300 | 0 |
| | | | | **Neg** | 1 | 99 |
| **Training 2** | 0.9457*** | 0.9673 | 0.8846*** | **Pos** | 356 | 15 |
| | | | | **Neg** | 12 | 115 |
| **visual examination** | 0.8025 | 1* | 0.5586 | **Pos** | 221 | 79 |
| | | | | **Neg** | 0 | 100 |

Abbreviation: Pos, positive samples; Neg, negative samples; PPV, positive predictive value; NPV, negative predictive value.

[a] PPV = numbers of true positives/(numbers of true positives+ numbers of false positives). The ideal value of the PPV for a perfect test will be 1, while the worst value will be zero.

[b] NPV = numbers of true negatives/(numbers of true negatives+ numbers of false negatives). The ideal value of the NPV for a perfect test will be 1, while the worst value will be zero

***: p-value <0.001

*: p-value <0.05

### Transfer learning model

Using transfer learning is one way to deal with small amount of training data. In our case, however, the results were not as good as we had expected. The architecture of our transfer learning model is taking whole convnet block of VGG19 [16], followed by four fully-connected layers. The accuracy of the transfer learning model is lower than the model without transfer learning as shown in S2 Table. We suspected that one reason is that the trainable parameters of the transfer learning model are actually much more than previous model.

## Discussion

In this study, we used two different training sets to train a deep CNN model which can recognize influenza-induced cytopathic effects. Both of them showed great sensitivity and specificity in recognition of influenza-induced cytopathic effects, whereas the Training 1 displayed slightly higher performance. The only difference between two trainings was the composition of training data, with Training 1 trained by 601 images and the Training 2 by 503, yet with more diverse data. The increment in number of training data was probably contributed by the bias of training where more sample number would provide more information and fewer sample number with less information may be less accurate. Therefore, Training 1 displayed better performance on sensitivity, even on specificity. Secondly, due to the higher amount of positive testing samples, an improved overall accuracy of Training 1 was observed, although Training 1 also tends to misjudge the images of non-influenza-infected samples as influenza-induced cytopathic effects. Nevertheless, as compared to Training 2, Training 1 could statistically significantly recognize the influenza-induced cytopathic effects at earlier time points. Hence, based on the aim for earlier recognition, we concluded that Training 1 is superior and is recommended for future applications.

In this study, we aimed to develop a model which can timely recognize the morphological changes induced by virus infection, which generally takes several days by manual observation in clinical practice. In our analysis, the accuracy for images taken at 28 hpi was 98.25% and 96.25%, respectively, for Trainings 1 and 2. We further examined whether our models could detect cytopathic effects at earlier time points, i.e. 16 hpi. At higher virus inoculation (0.5 M.O.

I.), both Trainings exhibited relatively comfortable recognition rates, 88.88% and 67.9%, respectively. At lower virus input (0.05 M.O.I.), the accuracy decreased to 15.92% and 5.3%, respectively. There was a slight gap in the recognition rate of the earlier experiment data. One of the reasons for the deficiency in the recognition of those pictures, especially 0.05 M.O.I., may be that the morphology of cells infected with influenza at a shorter infection period was too similar to those of the mock-infected cells (Fig 2A, 2E and 2F). However, at that stage, no significant difference could be even noticed by manual observation. In fact, in the clinical practice, some laboratories use R-mix to accelerate the process of virus identification. R-mix is a mixture of two cell lines for isolation of viruses. However, unlike manual observation, its results are determined by anti-virus specific antibody and immunofluorescent assay. In a recent study which compared the performance between R-mix and conventional culture, the rates of influenza A and B virus identification were 78% and 91%, respectively, by R-mix as compared to 25% and 39% using conventional culture [17]. Considering our accuracy for images taken at 28 hpi was 98.25% and 96.25%, respectively, for Trainings 1 and 2, our model is superior to R-mix, which has been shown to be better than the conventional culture.

The recognition rate of images taken at 40 hpi was slightly reduced to 87.46% and 77.33% for Trainings 1 and 2, respectively. The reduced recognition was likely contributed to the detachment of virus-infected cells at later time points, where more than 50% of cells developed cytopathic effects. However, the development of virus-induced cytopathic effects is a dynamic process and our model was designed as the screening machine for early recognition of influenza-infected samples. Therefore, the decreased accuracy of positive samples with late stage of influenza-induced cytopathic effects would not reduce the advantages of our model, since our model could identify the cytopathic effects in positive samples at earlier time points.

In this study, we used the PR8 virus strain to conduct the experiment. The PR8 virus strain is a laboratory reference strain for influenza study and the CPE induced by PR8 is representative to most influenza viruses. To examine that our model can recognize influenza CPE induced by different influenza virus strains, we collected about 120 image of MDCK cells infected by pandemic H1N1/09, H3N2, and influenza B at 16, 25, 28, and 40 hpi, respectively. Most results of recognition were as good as our testing data and experiment data, even better (S4 Table). The slight difference might be contributed by the different growth kinetics of influenza strains.

Compared to CNN models used in general, our models were relatively simple, consisting of only ten hidden layers. We did not choose a deeper model with the pretraining process because we believe the calculation would be more efficient with a non-complex model. Although we trained the model with a relative minute quantity of 1200 epochs, our model still showed remarkable accuracy in recognition of influenza-induced cytopathic effects. First, we only classified our data into two categories. For clinical diagnosis, only positive or negative readouts are required for final reports, rather than the numerical percentages where we believe a complex model would require more classification with more training epochs and more training data. Second, we utilized the original size of the photos, 1024x1360 pixels. Many reports described their model with a smaller size input [18,19], and some researches even adjusted the images to fit the pretraining model [5,6,8]. By downscaling the images, it may lead to loss of some features and in turns cause more difficulty in model training. The original size of the images is believed to provide more information to the model. To examine this possibility, we also tried to train the same model with a quarter size of the original pictures. Nevertheless, the training results were not prominent (S5 Table). It suggests that a smaller size of input does affect the complex of the models.

Considering the input size and patterns of influenza-induced cytopathic effects, we used larger sizes of convolutional filters and max-pooling kernels than common situations. Based

on the filer size of the first convolutional layer we used, it is estimated that four to nine cells can be covered, which should provide sufficient information to recognize the features of cytopathic effects. The increased kernel sizes of max-pooling can help to describe the differences regarding the location of images, considering the patterns of influenza-induced cytopathic effects might include the empty holes in the cell layer. These strategies in combination could reduce the complexity of the model without transfer learning, while maintaining the great performances. We also compared our results with those from a model applying transfer learning. The accuracy of the model without transfer learning was better by at least 20%. Therefore, we established a simple model with small amounts of training data, yet having great performance.

Our model has the advantage in shortening the screening time for virus isolation. Indeed, it is expected that a better discrepancy could be achieved at the later time point when a higher percentage of cytopathic effects develop. Therefore, one of the potential pitfalls for our model would be higher false positive ratios at earlier time point. If we assume the percentage of positive samples in the clinical laboratory is 30%, we can roughly estimate the false positive ratio. For cell culture at 25 hours post infection, the sensitivity of Training 1 was 100% and the specificity was 99%, which lead to a false positive ratio of 2.28%. The amount of misjudgment was acceptable because a subsequent immunofluorescence assay will be performed in the clinical diagnosis procedure to confirm the positivity of the observation.

In summary, we constructed a ten-layer convolutional neural network to differentiate influenza-induced cytopathic effects from normal MDCK cells. The model shows great performance both in accuracy and specificity with two trainings, although the recognition of initial cytopathic effect at early stages is not as impressive and can be further strengthened. Besides influenza virus, many other viruses can induce variant patterns of cytopathic effects on specific cells. We will try to train other models with the same structure to recognize different kinds of cytopathic effects. We also believe that our model structure could be applied on other similar imaging approaches, such as images of fluorescence assay and immunohistochemistry.

## Materials and methods

### Cells and viruses

MDCK cells (ATCC CCL-34, USA) were cultured in Modified Eagle's Medium (MEM; 61100–61, Gibco, USA) supplemented with 10% fetal bovine serum (FBS; 10082–147, Gibco, USA) and penicillin G sodium 100 units/mL, streptomycin sulfate 100 ug/mL and amphotericin B 250 ng/mL (antibiotic-antimycotic; 15240–062, Gibco, USA) at 37˚C with 5% CO2. The viruses used in this study included influenza laboratory strain PR8 virus [A/Puerto Rico/8/34 (H1N1)], pandemic H1N1 virus [A/California/07/2009 (H1N1)], H3N2 clinical isolate [A/Hong Kong/4801/2014 (H3N2)-like], influenza B clinical isolate [Victoria lineage], adenovirus, coxsackievirus B3, parainfluenza virus, respiratory syncytial virus (RSV), herpes simplex virus type 1 (HSV-1) and herpes simplex virus type 2 (HSV-2). Influenza virus and parainfluenza virus were amplified in MDCK cells, and the infected cells were maintained in the MEM with 2ug/mL tosylphenyl alanyl chloromethyl ketone (TPCK) trypsin (T1426, Sigma-Aldrich, USA). Other viruses were amplified in HEp2 or MK2 cells, and the infected cells were maintained in the MEM with 2% fetal bovine serum. The harvested virus supernatant was titrated by the plaque assay and stored at -80˚C for subsequent experiments.

### Virus infection

MDCK cells ($2x10^6$ cells/well) were infected with influenza virus, adenovirus, HSV-1, HSV-2, coxsackievirus B3, parainfluenza virus, and RSV for one hour before the addition of the MEM supplemented with 2ug/mL TPCK trypsin. The multiplicity of infection (M.O.I.) of influenza

virus used in this study was 0.01, 0.05, 0.1, 0.5, 1 and 2, respectively. The cytopathic effects were recorded by the Olympus IX71 microscope at 16 and 48 hours after infection.

## Datasets and taxonomy

The dataset was split into four sets, training set, validation set, testing set, and experiment set (including influenza experiment set and other viruses set). The cytopathic effects were monitored at 16 hours post infection. Six independent influenza virus infection experiments were conducted and the photos taken in the last two experiments were used as the testing set and the influenza experiment set. All of the training data (including training set and validation set), testing data (testing set) and influenza experiment data (influenza experiment set) included photos of uninfected MDCK cells and influenza-infected MDCK cells. The training, validation, and influenza experiment datasets classify photos into two categories. Photos taken in the well of uninfected MDCK cells were labeled as negative samples; the others with confirmed cytopathic effects were marked as positive samples. The images of other viruses set were all taken at 40 hours post infection, and they were not further labeled.

## Data collection

The images of cytopathic effects were recorded by the Olympus IX71 microscope. After the fields were selected, photos were taken continuously and refocused randomly. Because not all fields of influenza virus infected cells contained the cytopathic effects, we only took the fields with cytopathic effects as positive samples. The fields of uninfected cells were taken randomly as negative samples. For the independence between data sets, images of the training, testing set, and influenza experiment set were not taken from the same infection experiment.

To train the machine, we selected 601 photos, including 447 from influenza-infected MDCK cells and 154 from uninfected MDCK cells, respectively. After that, additional 400 pictures were used to test the model. Additionally, a total of 1190 images were taken as experiment set to evaluate the limit of the model, including 824 from influenza-infected MDCK cells and 366 from mock-infected MDCK cells. The image numbers for the other viruses set were 120. A total of six non-influenza viruses were included and twenty pictures from each virus infected cells were included for analysis. The output of other viruses set was negative. The detail would be revealed in the result section. If the non-influenza virus induced cytopathic effects on MDCK cells, these areas would be chosen for image taken. Otherwise, pictures were taken randomly. All of the original images were color with 1024 ×1360 pixels.

## Model architecture

Our deep convolutional neural network model consists of nine hidden layers, as shown in Fig 5. The first six layers are convolutional network and the following three layers are fully-connected network. The output layer is fed to a softmax generating two categorization labels.

We use three convolutional layers, and each is followed by a max-pooling layer. Stride factor s in each convolutional layer is one. The max-pooling layer is as large as kernel size. Each input image is a grayscale image of size $1024 \times 1360$. The first convolutional layer has 20 kernels of size $7 \times 7$ and following a max-pooling size of $10 \times 10$. The second convolutional layer filers the input from the previous layer with 25 kernels of size $5 \times 5$. Then, the second kernel size of max-pooling is $6 \times 6$. The third convolutional layer has 30 kernel of size $3 \times 3$ and following is a max-pooling layer of size $6 \times 6$. Then, each fully-connected layers are 100 neurons. The reason we choose somewhat large size of both filter and max-pooling kernel than traditional approach is because our input image is large. In the 3-layer fully-connected network, each has 100 neurons. Finally, there are two outputs and we apply softmax function to get the

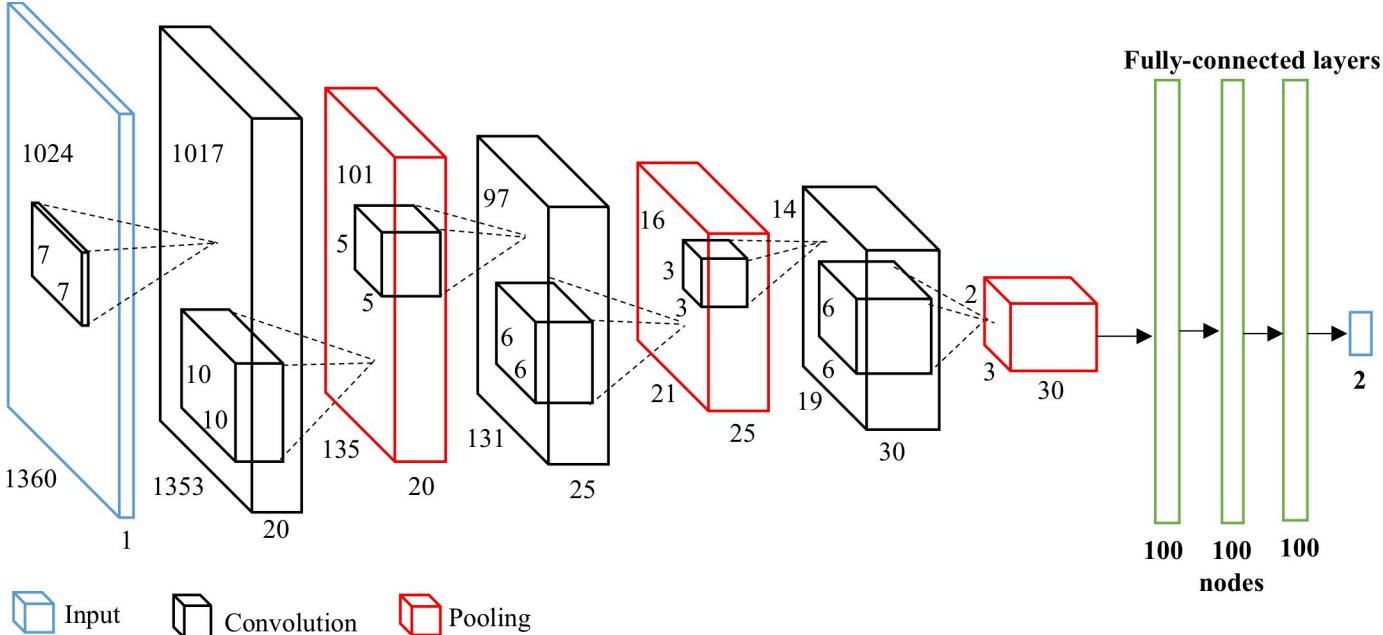

**Fig 5. Architecture of the Convolutional Neural Network model.** There are three convolutional layers with max pooling and four fully-connected layers. The numbers beside the input indicate the size of the image. The numbers of the convolutions and pooling are the size of output from previous layer. The numbers in the cube indicate the size of convolutional filter and pooling kernel. The bold numbers are the number of the neurons in the fully-connected layers.

final result. The code and data can be downloaded from https://github.com/b96404011/influenzaCPE_CNN.

## Training

Our CNN model was trained with back propagation. The weights of the CNN model are initialized randomly. All layers of the neural networks used the same global learning rate of 0.0001 because our model scale is small, only ten layers. We used Adam with the preset parameter value, namely, $\beta_1 = 0.9$, $\beta_2 = 0.999$, epsilon = 0, and decay = 0. We used TensorFlow deep learning framework to train, validate, and test our CNN model.

## Transfer learning

VGG19 [16] is a popular convolutional neural network model with 19 layers which is often used in transfer learning model. We took the whole convolutional part of VGG19 and connected with three fully-connected layers, each of 100 neurons. We upsample input images to 1024x1360x3 to fit the input of VGG19. The weights of the network from VGG19 are fixed (from the original VGG19). Only weights of newly added fully-connected layers are trainable. The other statuses are the same as our own model. We used Adam as the optimizer to train the neural network.

## Statistical analysis

All statistical analyses were performed using SPSS software version 16.0 (SPSS Inc., Chicago, IL). Categorical variables were compared using $\chi^2$ or Fisher's exact test. A $p < 0.05$ was considered statistically significant.

## Supporting information

**S1 Table. Information Table for the Training 1 Initial Testing Data Set.**
(DOC)

**S2 Table. Transfer Learning Model Comparison of Training 1 and Training 2.**
(DOC)

**S3 Table. Comparison of Training 1 and Training 2 with 1200 epochs weights on other viruses infected images at earlier time point.**
(DOC)

**S4 Table. Comparison of Training 1 and Training 2 with 1200 epochs weights on other strains of influenza virus data.**
(DOC)

**S5 Table. CNN Model with quarter size images.**
(DOC)

## Author Contributions

**Conceptualization:** Sui-Yuan Chang.

**Formal analysis:** Ting-En Wang.

**Investigation:** Tai-Ling Chao, Pi-Han Lin, Yen-Lung Tsai.

**Methodology:** Ting-En Wang, Tai-Ling Chao, Hsin-Tsuen Tsai, Pi-Han Lin.

**Supervision:** Sui-Yuan Chang.

**Writing – original draft:** Ting-En Wang.

**Writing – review & editing:** Yen-Lung Tsai, Sui-Yuan Chang.

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
