## [Decision Letter · Decision Letter 0]

6 Jan 2020

Dear Dr Chang,

Thank you very much for submitting your manuscript 'Differentiation of Cytopathic Effects (CPE) Induced by Influenza Virus Infection Using Deep Convolutional Neural Networks (CNN)' for review by PLOS Computational Biology. Your manuscript has been fully evaluated by the PLOS Computational Biology editorial team and in this case also by independent peer reviewers. The reviewers appreciated the attention to an important problem, but raised some substantial concerns about the manuscript as it currently stands. While your manuscript cannot be accepted in its present form, we are willing to consider a revised version in which the issues raised by the reviewers have been adequately addressed. We cannot, of course, promise publication at that time.

Sincerely,

Amber M Smith

Guest Editor

PLOS Computational Biology

Rob De Boer

Deputy Editor

PLOS Computational Biology

[LINK]

Reviewer's Responses to Questions

**Comments to the Authors:**

Reviewer #1: The authors Chang et al. described a CNN that classifies samples into two categories (infected with flu vs not infected with flu) based on images of cells cultured with sample after various times. Two training/validation sets have been constructed and results were similar. The results appear to be robust and have clinical application.

Major issues:

The training data set (images, and their labels) are not provided.

It would be better if the code and trained model is also available.

When images of cells infected with other viruses are used as the input, what is the output? The data in Table 4 indicated that for RSV, training 2, 1200 epochs, accuracy is 1 but the best one is 0.35. What does the column "best one" indicate? Why is the best one is 0.35 and 1200 epoc is 1? Should the best one be better than 1200 epochs (most of the rows are but not always)?

Based on the images provide for the other virus-infected cells, some of them have apparent changes in cell morphology. If they are not classified as influenza damage, are they classified as no damage? (because the CNN only has two outputs).

The manuscript can use some better organization regarding the introduction of training1 vs. training2. Training 2 data appeared before they were introduced, I suspect this was a response to a prior review of some sort.

Minor issues

The manuscript could use some editing of the English Language.

For example:

Page 10: line 145: were differentiated correctly: Do the author mean identified?

Page 19: line 220: resigned training data? What is "resigned" referring to?

Reviewer #2: The manuscript by Want et al. describes a deep convolutional neural network (CNN) approach to detect influenza induced cytopathic effects from microscopic images of influenza infected MDCK cells. The aim is to develop a fast and accurate tool for rapid influenza diagnoses in clinics. Deep learning approaches have been proven to be extremely powerful for identifying objects of interests from images in the past decade. Thus, the approach to the problem was well chosen in the study. The authors showed that trained model can achieve a very high percentage of correct recognition (99.75%). Although the results look convincing, my enthusiasm about the study is dampened by a couple of major concerns and some minor concerns.

First of all, the images used to train the model are images of cells infected by a single virus strain (PR8). While the authors tested the model against a variety of other non-influenza viruses, I think it is critical/essential to test how the model works with other influenza strains. This is because we usually do not know the virus strain that a person is infected with as a priori. The model is useful only when it can correctly predict the presence of influenza virus irrespective of the strain a cell is infected with.

Second, it is not clear to me how this method is compared with other method in terms of accuracy and time needed for diagnoses. Although the authors discussed and compared the performance of their model with the performance of R-mix, the comparison does not seem to be fair, because the tests were done using different sets of data and images collected from different experiment with different protocols. Furthermore, the CNN model may be an automation method and a useful alternative to visual examination, for example by a medical technologist. However, it is not clear to me how much (quantitative) benefit one gains as compared to visual examination, e.g. how much more accurate the CNN model performs compared to visual examination and how much time it saves. Overall, the advantages and significance of this approach in terms of clinical use is not clearly discussed and highlighted. This really limit the enthusiasm about the proposed approach.

Third, in Fig 4, the authors used the model to test against non-influenza viruses and concluded that ’no cytopathic effect was observed in MDCK cells infected by other viruses’. It is not clear to me what analysis was exactly performed. Did the author use the model trained on influenza infected and mock MDCK cells to classify images of cells infected by non-influenza viruses; or the authors retrained the model including the images of cells infected by non-influenza viruses? In the images with other virus infections, is that indeed ‘there is no cytopathic effects’; or other viruses cause cytopathic effects that are different from influenza? I also notice that the images were all taken at 40phi (for other viruses). What about other time points?

Some minor points below:

Lines 186- 201, it is not clear to me how the accuracy of prediction of images taken at different time points were done. Did the authors retrain the model; or just used the model trained on the Training dataset described in the earlier text?

Line 194-195, Can the authors test why the accuracy drops at 40hpi? It is not sufficient to speculate the cause.

Line 198, typo ‘act’ -> ‘fact’?

**Have all data underlying the figures and results presented in the manuscript been provided?**

Reviewer #1: No: The training data set (images and labels) are not provided.

Reviewer #2: Yes

PLOS authors have the option to publish the peer review history of their article (what does this mean?). If published, this will include your full peer review and any attached files.

Reviewer #1: No

Reviewer #2: No

---

## [Decision Letter · Decision Letter 1]

26 Mar 2020

Dear Dr. Chang,

Thank you very much for submitting your manuscript "Differentiation of Cytopathic Effects (CPE) Induced by Influenza Virus Infection Using Deep Convolutional Neural Networks (CNN)" for consideration at PLOS Computational Biology. As with all papers reviewed by the journal, your manuscript was reviewed by members of the editorial board and by several independent reviewers. The reviewers appreciated the attention to an important topic. Based on the reviews, we are likely to accept this manuscript for publication, providing that you modify the manuscript according to the review recommendations. 

Sincerely,

Amber M Smith

Guest Editor

PLOS Computational Biology

Rob De Boer

Deputy Editor

PLOS Computational Biology

[LINK]

Reviewer's Responses to Questions

**Comments to the Authors:**

Reviewer #1: The authors need to incorporate the reply to reviewer's question in the revised manuscript. Most of the answers to Reviewer 1 were not reflected in the revision. The authors stated that they changed "best one" to "saved weights". But in table 2, they still used "best one". Also although the code is provided in github (much appreciated), the data is provided through a link. The link time out when I tried to download the data.

Reviewer #2: All my concerns are appropriately addressed. I appreciate the author's efforts to collected additional data (images) of cells infected by other strains. This added a lot of robustness to the model.

**Have all data underlying the figures and results presented in the manuscript been provided?**

Reviewer #1: No: As stated, the data were not available.

Reviewer #2: Yes

PLOS authors have the option to publish the peer review history of their article (what does this mean?). If published, this will include your full peer review and any attached files.

Reviewer #1: No

Reviewer #2: Yes: Ruian Ke
---

## [Editor Report · Decision Letter 2]

16 Apr 2020

Dear Dr. Chang,

We are pleased to inform you that your manuscript 'Differentiation of Cytopathic Effects (CPE) Induced by Influenza Virus Infection Using Deep Convolutional Neural Networks (CNN)' has been provisionally accepted for publication in PLOS Computational Biology.

Best regards,

Amber M Smith

Guest Editor

PLOS Computational Biology

Rob De Boer

Deputy Editor

PLOS Computational Biology

---

## [Editor Report · Acceptance letter]

30 Apr 2020

PCOMPBIOL-D-19-01629R2 

Differentiation of Cytopathic Effects (CPE) Induced by Influenza Virus Infection Using Deep Convolutional Neural Networks (CNN)

Dear Dr Chang,

I am pleased to inform you that your manuscript has been formally accepted for publication in PLOS Computational Biology. Your manuscript is now with our production department and you will be notified of the publication date in due course.

With kind regards,

Laura Mallard
